# On Sampling from the Gibbs Distribution with Random Maximum A-Posteriori Perturbations

**Tamir Hazan**
University of Haifa

**Subhransu Maji**
TTI Chicago

**Tommi Jaakkola**
CSAIL, MIT

## Abstract

In this paper we describe how MAP inference can be used to sample efficiently from Gibbs distributions. Specifically, we provide means for drawing either approximate or unbiased samples from Gibbs' distributions by introducing low dimensional perturbations and solving the corresponding MAP assignments. Our approach also leads to new ways to derive lower bounds on partition functions. We demonstrate empirically that our method excels in the typical "high signal - high coupling" regime. The setting results in ragged energy landscapes that are challenging for alternative approaches to sampling and/or lower bounds.

## 1   Introduction

Inference in complex models drives much of the research in machine learning applications, from computer vision, natural language processing, to computational biology. Examples include scene understanding, parsing, or protein design. The inference problem in such cases involves finding likely structures, whether objects, parsers, or molecular arrangements. Each structure corresponds to an assignment of values to random variables and the likelihood of an assignment is based on defining potential functions in a Gibbs distribution. Usually, it is feasible to find only the most likely or maximum a-posteriori (MAP) assignment (structure) rather than sampling from the full Gibbs distribution. Substantial effort has gone into developing algorithms for recovering MAP assignments, either based on specific structural restrictions such as super-modularity [2] or by devising cutting-planes based methods on linear programming relaxations [19, 24]. However, MAP inference is limited when there are other likely assignments.

Our work seeks to leverage MAP inference so as to sample efficiently from the full Gibbs distribution. Specifically, we aim to draw either approximate or unbiased samples from Gibbs distributions by introducing low dimensional perturbations in the potential functions and solving the corresponding MAP assignments. Connections between random MAP perturbations and Gibbs distributions have been explored before. Recently [17, 21] defined probability models that are based on low dimensional perturbations, and empirically tied them to Gibbs distributions. [5] augmented these results by providing bounds on the partition function in terms of random MAP perturbations.

In this work we build on these results to construct an efficient sampler for the Gibbs distribution, also deriving new lower bounds on the partition function. Our approach excels in regimes where there are several but not exponentially many prominent assignments. In such ragged energy landscapes classical methods for the Gibbs distribution such as Gibbs sampling and Markov chain Monte Carlo methods, remain computationally expensive [3, 25].

## 2   Background

Statistical inference problems involve reasoning about the states of discrete variables whose configurations (assignments of values) specify the discrete structures of interest. We assume that the

models are parameterized by real valued potentials $\theta(x) = \theta(x_1, ..., x_n) < \infty$ defined over a discrete product space $X = X_1 \times \cdots \times X_n$. The effective domain is implicitly defined through $\theta(x)$ via exclusions $\theta(x) = -\infty$ whenever $x \notin dom(\theta)$. The real valued potential functions are mapped to the probability scale via the Gibbs' distribution:

$$p(x_1, ..., x_n) = \frac{1}{Z} \exp(\theta(x_1, ..., x_n)), \text{ where } Z = \sum_{x_1, ..., x_n} \exp(\theta(x_1, ..., x_n)). \quad (1)$$

The normalization constant $Z$ is called the partition function. The feasibility of using the distribution for prediction, including sampling from it, is inherently tied to the ability to evaluate the partition function, i.e., the ability to sum over the discrete structures being modeled. In general, such counting problems are often hard, in #P.

A slightly easier problem is that of finding the most likely assignment of values to variables, also known as the maximum a-posterior (MAP) prediction.

$$\text{(MAP)} \qquad \arg \max_{x_1, ..., y_n} \theta(x_1, ..., x_n) \qquad (2)$$

Recent advances in optimization theory have been translated to successful algorithms for solving such MAP problems in many cases of practical interest. Although the MAP prediction problem is still NP-hard in general, it is often simpler than sampling from the Gibbs distribution.

Our approach is based on representations of the Gibbs distribution and the partition function using extreme value statistics of linearly perturbed potential functions. Let $\{\gamma(x)\}_{x \in X}$ be a collection of random variables with zero mean, and consider random potential functions of the form $\theta(x) + \gamma(x)$. Analytic expressions for the statistics of a randomized MAP predictor, $\hat{x} \in \operatorname{argmax}_x \{\theta(x) + \gamma(x)\}$, can be derived for general discrete sets, whenever independent and identically distributed (i.i.d.) random perturbations are applied for every assignment $x \in X$. Specifically, when the random perturbations follow the Gumbel distribution (cf. [12]), we obtain the following result.

**Theorem 1.** *([4], see also [17, 5]) Let $\{\gamma(x)\}_{x \in X}$ be a collection of i.i.d. random variables, each following the Gumbel distribution with zero mean, whose cumulative distribution function is $F(t) = \exp(-\exp(-(t + c)))$, where $c$ is the Euler constant. Then*

$$\log Z = E_\gamma \left[ \max_{x \in X} \{\theta(x) + \gamma(x)\} \right].$$

$$\frac{1}{Z} \exp(\theta(\hat{x})) = P_\gamma \left[ \hat{x} \in \arg \max_{x \in X} \{\theta(x) + \gamma(x)\} \right].$$

The max-stability of the Gumbel distribution provides a straight forward approach to generate unbiased samples from the Gibbs distribution as well as to approximate the partition function by a sample mean of random MAP perturbation. Assume we sample $j = 1, ..., m$ independent predictions $\max_x \{\theta(x) + \gamma_j(x)\}$, then every maximal argument is an unbiased sample from the Gibbs distribution. Moreover, the randomized MAP predictions $\max_x \{\theta(x) + \gamma_j(x)\}$ are independent and follow the Gumbel distribution, whose variance is $\pi^2/6$. Therefore Chebyshev's inequality dictates, for every $t, m$

$$Pr_\gamma \left[ \left| \frac{1}{m} \sum_{j=1}^{m} \max_x \{\theta(x) + \gamma_j(x)\} - \log Z \right| \geq \epsilon \right] \leq \frac{\pi}{6m\epsilon^2} \qquad (3)$$

In general each $x = (x_1, ..., x_n)$ represents an assignment to $n$ variables. Theorem 1 suggests to introduce an independent perturbation $\gamma(x)$ for each such $n-$dimensional assignment $x \in X$. The complexity of inference and learning in this setting would be exponential in $n$. In our work we propose to investigate low dimensional random perturbations as the main tool to efficiently (approximate) sampling from the Gibbs distribution.

## 3 Probable approximate samples from the Gibbs distribution

Sampling from the Gibbs distribution is inherently tied to estimating the partition function. Markov properties that simplify the distribution also decompose the computation of the partition function.

For example, assume a graphical model with potential functions associated with subsets of variables $\alpha \subset \{1, ..., n\}$ so that $\theta(x) = \sum_{\alpha \in \mathcal{A}} \theta_\alpha(x_\alpha)$. Assume that the subsets are disjoint except for their common intersection $\beta = \cap_{\alpha \in \mathcal{A}}$. This separation implies that the partition function can be computed in lower dimensional pieces

$$Z = \sum_{x_\beta} \prod_{\alpha \in \mathcal{A}} \Big( \sum_{x_\alpha \setminus x_\beta} \exp(\theta_\alpha(x_\alpha)) \Big)$$

As a result, the computation is exponential only in the size of the subsets $\alpha \in \mathcal{A}$. Thus, we can also estimate the partition function with lower dimensional random MAP perturbations, $E_\gamma[\max_{x_\alpha \setminus x_\beta} \{\theta_\alpha(x_\alpha) + \gamma_\alpha(x_\alpha)\}]$. The random perturbation are now required only for each assignment of values to the variables within the subsets $\alpha \in \mathcal{A}$ rather than the set of all variables.

We approximate such partition functions with low dimensional perturbations and their averages. The overall computation is cast in a single MAP problem using an extended representation of potential functions by replicating variables.

**Lemma 1.** *Let $\mathcal{A}$ be subsets of variables that are separated by their joint intersection $\beta = \cap_{\alpha \in \mathcal{A}} \alpha$. We create multiple copies of $x_\alpha$, namely $\hat{x}_\alpha = (x_{\alpha, j_\alpha})_{j_\alpha = 1, ..., m_\alpha}$, and define the extended potential function $\hat{\theta}_\alpha(\hat{x}_\alpha) = \sum_{j_\alpha = 1}^{m_\alpha} \theta_\alpha(x_{\alpha, j_\alpha})/m_\alpha$. We also define the extended perturbation model $\hat{\gamma}_\alpha(\hat{x}_\alpha) = \sum_{j_\alpha = 1}^{m_\alpha} \gamma_{\alpha, j_\alpha}(x_{\alpha, j_\alpha})/m_\alpha$, where each $\gamma_{\alpha, j_\alpha}(x_{\alpha, j_\alpha})$ is independent and distributed according to the Gumbel distribution with zero mean. Then, for every $x_\beta$, with probability at least $1 - \sum_{\alpha \in \mathcal{A}} \frac{\pi^2}{6 m_\alpha \epsilon^2}$*

$$\Big| \max_{\hat{x} \setminus x_\beta} \Big\{ \sum_{\alpha \in \mathcal{A}} \hat{\theta}_\alpha(\hat{x}_\alpha) + \sum_{\alpha \in \mathcal{A}} \hat{\gamma}_\alpha(\hat{x}_\alpha) \Big\} - \sum_{\alpha \in \mathcal{A}} \log \Big( \sum_{x_\alpha \setminus x_\beta} \exp(\theta_\alpha(x_\alpha)) \Big) \Big| \le \epsilon |\mathcal{A}|$$

**Proof:** Equation (3) implies that for every $x_\beta$ with probability at most $\pi^2/6 m_\alpha \epsilon^2$ holds

$$\Big| \frac{1}{m_\alpha} \sum_{j_\alpha = 1}^{m_\alpha} \max_{x_\alpha \setminus x_\beta} \{\theta_\alpha(x_\alpha) + \gamma_{\alpha, j_\alpha}(x_\alpha)\} - \log \Big( \sum_{x_\alpha \setminus x_\beta} \exp(\theta_\alpha(x_\alpha)) \Big) \Big| \le \epsilon.$$

To compute the sampled average with a single max-operation we introduce the multiple copies $\hat{x}_\alpha = (x_{\alpha, j_\alpha})_{j_\alpha = 1, ..., m_\alpha}$ thus $\sum_{j_\alpha = 1}^{m_\alpha} \max_{x_\alpha \setminus x_\beta} \{\theta_\alpha(x_\alpha) + \gamma_{\alpha, j_\alpha}(x_\alpha)\} = \max_{x_{\alpha, j_\alpha} \setminus x_\beta} \sum_{j=1}^{m} \{\theta_\alpha(x_{\alpha, j_\alpha}) + \gamma_{\alpha, j_\alpha}(x_{\alpha, j_\alpha})\}$. By the union bound it holds for every $\alpha \in \mathcal{A}$ simultaneously with probability at least $1 - \sum_{\alpha \in \mathcal{A}} \pi^2/6 m_\alpha \epsilon^2$. Since $x_\beta$ is fixed for every $\alpha \in \mathcal{A}$ the maximizations are done independently across subsets in $\hat{x} \setminus x_\beta$, where $\hat{x}$ is the concatenation of all $\hat{x}_\alpha$, and

$$\sum_{\alpha \in \mathcal{A}} \max_{\hat{x}_\alpha \setminus x_\beta} \sum_{j_\alpha = 1}^{m_\alpha} \Big\{ \theta_\alpha(x_{\alpha, j_\alpha}) + \gamma_{\alpha, j_\alpha}(x_{\alpha, j_\alpha}) \Big\} = \max_{\hat{x} \setminus x_\beta} \sum_{j_\alpha = 1}^{m_\alpha} \Big\{ \sum_{\alpha \in \mathcal{A}} \theta_\alpha(x_{\alpha, j_\alpha}) + \sum_{\alpha \in \mathcal{A}} \gamma_{\alpha, j_\alpha}(x_{\alpha, j_\alpha}) \Big\}.$$

The proof then follows from the triangle inequality. □

Whenever the graphical model has no cycles we can iteratively apply the separation properties without increasing the computational complexity of perturbations. Thus we may randomly perturb the subsets of potentials in the graph. For notational simplicity we describe our approximate sampling scheme for pairwise interactions $\alpha = (i, j)$ although it holds for general graphical models without cycles:

**Theorem 2.** *Let $\theta(x) = \sum_{i \in V} \theta_i(x_i) + \sum_{i,j \in E} \theta_{i,j}(x_i, x_j)$ be a graphical model without cycles, and let $p(x)$ be the Gibbs distribution defined in Equation (1). Let $\hat{\theta}(x) = \sum_{k_i = 1}^{m_i} \theta(x_{1, k_1}, ..., x_{n, k_n})/\prod_i m_i$, and $\hat{\gamma}_{i,j}(x_i, x_j) = \sum_{k_i, k_j = 1}^{m_i, m_j} \gamma_{i,j,k_i,k_j}(x_{i,k_i}, x_{j,k_j})/m_i m_j$ where each perturbation is independent and distributed according to the Gumbel distribution with zero mean. Then, for every edge $(r, s)$ while $m_r = m_s = 1$ (i.e., they have no multiple copies) there holds with probability at least $1 - \sum_{i=1}^{n} \pi^2 c/6 m_i \epsilon^2$, where $c = \max_i |X_i|$*

$$\Big| \log \Big( P_\gamma \Big[ x_r, x_s \in \arg\max_{\hat{x}} \Big\{ \hat{\theta}(x) + \sum_{i,j \in E} \hat{\gamma}_{i,j}(x_i, x_j) \Big\} \Big] \Big) - \log \Big( \sum_{x \setminus x_r, x_s} p(x) \Big) \Big| \le \epsilon n$$

**Proof:** Theorem 1 implies that we sample $(x_r, x_s)$ approximately from the Gibbs distribution marginal probabilities with a max-operation, if we approximate $\sum_{x \setminus \{x_r, x_s\}} \exp(\theta(x))$. Using graph separation (or equivalently the Markov property) it suffices to approximate the partial partition function over the disjoint subtrees $T_r, T_s$ that originate from $r, s$ respectively. Lemma 1 describes this case for a directed tree with a single parent. We use this by induction on the parents on these directed trees, noticing that graph separation guarantees: the statistics of Lemma 1 hold uniformly for every assignment of the parent's non-descendants as well; the optimal assignments in Lemma 1 are chosen independently for every child for every assignment of the parent's non-descendants label. □

Our approximated sampling procedure expands the graphical model, creating layers of the original graph, while connecting edges between vertices in the different layers if an edge exists in the original graph. We use graph separations (Markov properties) to guarantee that the number of added layers is polynomial in $n$, while we approach arbitrarily close to the Gibbs distribution. This construction preserves the structure of the original graph, in particular, whenever the original graph has no cycles, the expanded graph does not have cycles as well. In the experiments we show that this probability model approximates well the Gibbs distribution for graphical models with many cycles.

## 4 Unbiased sampling using sequential bounds on the partition function

In the following we describe how to use random MAP perturbations to generate unbiased samples from the Gibbs distribution. Sampling from the Gibbs distribution is inherently tied to estimating the partition function. Assume we could have compute the partition function exactly, then we could have sample from the Gibbs distribution sequentially: for every dimension we sample $x_i$ with probability which is proportional to $\sum_{x_{i+1},...,x_n} \exp(\theta(x))$. Unfortunately, approximations to the partition function, as described in Section 3, cannot provide a sequential procedure that would generate unbiased samples from the full Gibbs distribution. Instead, we construct a family of self-reducible upper bounds which imitate the partition function behavior, namely bound the summation over its exponentiations. These upper bounds extend the one in [5] when restricted to local perturbations.

**Lemma 2.** *Let $\{\gamma_i(x_i)\}$ be a collection of i.i.d. random variables, each following the Gumbel distribution with zero mean. Then for every $j = 1, ..., n$ and every $x_1, ..., x_{j-1}$ holds*

$$\sum_{x_j} \exp \left( E_\gamma \left[ \max_{x_{j+1},...,x_n} \{\theta(x) + \sum_{i=j+1}^{n} \gamma_i(x_i)\} \right] \right) \leq \exp \left( E_\gamma \left[ \max_{x_j,...,x_n} \{\theta(x) + \sum_{i=j}^{n} \gamma_i(x_i)\} \right] \right)$$

*In particular, for $j = n$ holds $\sum_{x_n} \exp(\theta(x)) = \exp \left( E_{\gamma_n(x_n)} \left[ \max_{x_j,...,x_n} \{\theta(x) + \gamma_n(x_n)\} \right] \right)$.*

**Proof:** The result is an application of the expectation-optimization interpretation of the partition function in Theorem 1. The left hand side equals to $E_{\gamma_j} \left[ \max_{x_j} E_{\gamma_{j+1},...,\gamma_n} \left[ \max_{x_{j+1},...,x_n} \{\theta(x) + \sum_{i=j}^{n} \gamma_i(x_i)\} \right] \right]$, while the right hand side is attained by alternating the maximization with respect to $x_j$ with the expectation of $\gamma_{j+1}, ..., \gamma_n$. The proof then follows by taking the exponent. □

We use these upper bounds for every dimension $i = 1, ..., n$ to sample from a probability distribution that follows a summation over exponential functions, with a discrepancy that is described by the upper bound. This is formalized below in Algorithm 1

---

**Algorithm 1** Unbiased sampling from Gibbs distribution using randomized prediction

Iterate over $j = 1, ..., n$, while keeping fixed $x_1, ..., x_{j-1}$. Set

1. $p_j(x_j) = \frac{\exp \left( E_\gamma \left[ \max_{x_{j+1},...,x_n} \{\theta(x) + \sum_{i=j+1}^{n} \gamma_i(x_i)\} \right] \right)}{\exp \left( E_\gamma \left[ \max_{x_j,...,x_n} \{\theta(x) + \sum_{i=j}^{n} \gamma_i(x_i)\} \right] \right)}$.

2. $p_j(r) = 1 - \sum_{x_j} p(x_j)$

3. Sample an element according to $p_j(\cdot)$. If $r$ is sampled then reject and restart with $j = 1$. Otherwise, fix the sampled element $x_j$ and continue the iterations.

Output: $x_1, ..., x_n$

---

When we reject the discrepancy, the probability we accept a configuration $x$ is the product of probabilities in all rounds. Since these upper bounds are self-reducible, i.e., for every dimension $i$ we

are using the same quantities that were computed in the previous dimensions $1, ..., i-1$, we are sampling an accepted configuration proportionally to $\exp(\theta(x))$, the full Gibbs distribution.

**Theorem 3.** *Let $p(x)$ be the Gibbs distribution, defined in Equation (1) and let $\{\gamma_i(x_i)\}$ be a collection of i.i.d. random variables following the Gumbel distribution with zero mean. Then whenever Algorithm 1 accepts, it produces a configuration $(x_1, ..., x_n)$ according to the Gibbs distribution*

$$P\Big[\text{Algorithm 1 outputs } x \;\big|\; \text{Algorithm 1 accepts}\Big] = p(x).$$

**Proof:** The probability of sampling a configuration $(x_1, ..., x_n)$ without rejecting is

$$\prod_{j=1}^{n} \frac{\exp\big(E_\gamma\big[\max_{x_{j+1},...,x_n} \{\theta(x) + \sum_{i=j+1}^{n} \gamma_i(x_i)\}\big]\big)}{\exp\big(E_\gamma\big[\max_{x_j,...,x_n} \{\theta(x) + \sum_{i=j}^{n} \gamma_i(x_i)\}\big]\big)} = \frac{\exp(\theta(x))}{\exp\big(E_\gamma\big[\max_{x_1,...,x_n} \{\theta(x) + \sum_{i=1}^{n} \gamma_i(x_i)\}\big]\big)}.$$

The probability of sampling without rejecting is thus the sum of this probability over all configuration, i.e., $P[\text{Algorithm 1 accepts}] = Z\big/\exp\big(E_\gamma\big[\max_{x_1,...,x_n}\{\theta(x) + \sum_{i=1}^{n}\gamma_i(x_i)\}\big]\big)$. Therefore conditioned on accepting a configuration, it is produced according to the Gibbs distribution. $\square$.

Acceptance/rejection follows the geometric distribution, therefore the sampling procedure rejects $k$ times with probability $(1 - P[\text{Algorithm 1 accepts}])^k$. The running time of our Gibbs sampler is determined by the average number of rejections $1/P[\text{Algorithm 1 accepts}]$. Interestingly, this average is the quality of the partition upper bound presented in [5]. To augment this result we investigate in the next section efficiently computable lower bounds to the partition function, that are based on random MAP perturbations. These lower bounds provide a way to efficiently determine the computational complexity for sampling from the Gibbs distribution for a given potential function.

## 5 Lower bounds on the partition function

The realization of the partition function as expectation-optimization pair in Theorem 1 provides efficiently computable lower bounds on the partition function. Intuitively, these bounds correspond to moving expectations (or summations) inside the maximization operations. In the following we present two lower bounds that are derived along these lines, the first holds in expectation and the second holds in probability.

**Corollary 1.** *Consider a family of subsets $\alpha \in \mathcal{A}$ and let $x_\alpha$ be a set of variables $\{x_i\}_{i \in \alpha}$ restricted to the indexes in $\alpha$. Assume that the random variables $\gamma_\alpha(x_\alpha)$ are i.i.d. according to the Gumbel distribution with zero mean, for every $\alpha, x_\alpha$. Then*

$$\forall \alpha \in \mathcal{A} \quad \log Z \geq E_\gamma \Big[ \max_x \big\{ \theta(x) + \gamma_\alpha(x_\alpha) \big\} \Big].$$

*In particular,* $\log Z \geq E_\gamma\Big[\max_x \big\{\theta(x) + \frac{1}{|\mathcal{A}|}\sum_{\alpha \in \mathcal{A}} \gamma_\alpha(x_\alpha)\big\}\Big]$.

**Proof:** Let $\bar{\alpha} = \{1, ..., n\} \setminus \alpha$ then $Z = \sum_{x_\alpha} \sum_{x_{\bar{\alpha}}} \exp(\theta(x)) \geq \sum_{x_\alpha} \max_{x_{\bar{\alpha}}} \exp(\theta(x))$. The first result is derived by swapping the maximization with the exponent, and applying Theorem 1. The second result is attained while averaging these lower bounds $\log Z \geq \sum_{\alpha \in \mathcal{A}} \frac{1}{|\mathcal{A}|} E_\gamma[\max_x\{\theta(x) + \gamma_\alpha(x_\alpha)\}]$, and by moving the summation inside the maximization operation. $\square$

The expected lower bound requires to invoke a MAP solver multiple times. Although this expectation may be estimated with a single MAP execution, the variance of this random MAP prediction is around $\sqrt{n}$. We suggest to recursively use Lemma 1 to lower bound the partition function with a single MAP operation in probability.

**Corollary 2.** *Let $\theta(x)$ be a potential function over $x = (x_1, ..., x_n)$. We create multiple copies of $x_i$, namely $x_{i,k_i}$ for $k_i = 1, ..., m_i$, and define the extended potential function $\hat{\theta}(x) = \sum_{k_i=1}^{m_i} \theta(x_{1,k_1}, ..., x_{n,k_n})/\prod m_i$. We define the extended perturbation model $\hat{\gamma}_i(x_i) = \sum_{k_i=1}^{m_i} \gamma_{i,k_i}(x_{i,k_i})/m_i$ where each perturbation is independent and distributed according to the Gumbel distribution with zero mean. Then, with probability at least $1 - \sum_{i=1}^{n} \pi^2 |dom(\theta)|/6m_i \epsilon^2$ holds $\log Z \geq \max_{\hat{x}}\{\hat{\theta}(x) + \sum_{i=1}^{n} \hat{\gamma}_i(x_i)\} - \epsilon n$*

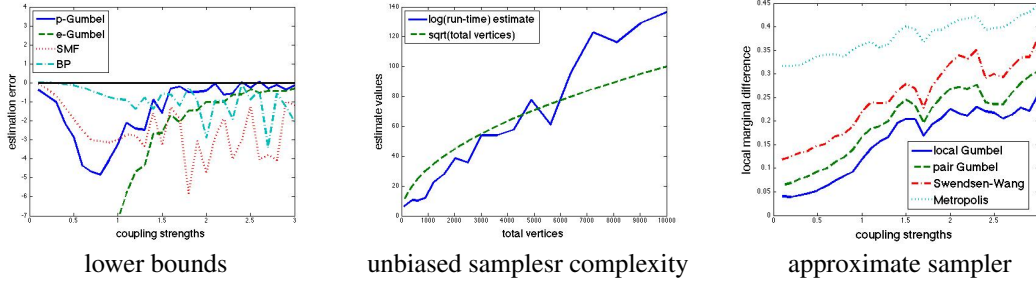

| lower bounds | unbiased samplesr complexity | approximate sampler |

Figure 1: *Left: comparing our expected lower and probable lower bounds with structured mean-field and belief propagation on attractive models with high signal and varying coupling strength. Middle: estimating our unbiased sampling procedure complexity on spin glass models of varying sizes. Right: Comparing our approximate sampling procedure on attractive models with high signal.*

**Proof:** We estimate the expectation-optimization value of the log-partition function iteratively for every dimension, while replacing each expectation with its sampled average, as described in Lemma 1. Our result holds for every potential function, thus the statistics in each recursion hold uniformly for every $x$ with probability at least $1 - \pi^2 |dom(\theta)|/6m_i\epsilon^2$. We then move the averages inside the maximization operation, thus lower bounding the $\epsilon n-$approximation of the partition function. $\square$

The probable lower bound that we provide does not assume graph separations thus the statistical guarantees are worse than the ones presented in the approximation scheme of Theorem 2. Also, since we are seeking for lower bound, we are able relax our optimization requirements and thus to use vertex based random perturbations $\gamma_i(x_i)$. This is an important difference that makes this lower bound widely applicable and very efficient.

## 6  Experiments

We evaluated our approach on spin glass models $\theta(x) = \sum_{i \in V} \theta_i x_i + \sum_{(i,j) \in E} \theta_{i,j} x_i x_j$. where $x_i \in \{-1, 1\}$. Each spin has a local field parameter $\theta_i$, sampled uniformly from $[-1, 1]$. The spins interact in a grid shaped graphical model with couplings $\theta_{i,j}$, sampled uniformly from $[0, c]$. Whenever the coupling parameters are positive the model is called attractive as adjacent variables give higher values to positively correlated configurations. Attractive models are computationally appealing as their MAP predictions can be computed efficiently by the graph-cut algorithm [2].

We begin by evaluating our lower bounds, presented in Section 5, on $10 \times 10$ spin glass models. Corollary 1 presents a lower bound that holds in expectation. We evaluated these lower bounds while perturbing the local potentials with $\gamma_i(x_i)$. Corollary 2 presents a lower bound that holds in probability and requires only a single MAP prediction on an expanded model. We evaluate the probable bound by expanding the model to $1000 \times 1000$ grids, ignoring the discrepancy $\epsilon$. For both the expected lower bound and the probable lower bound we used graph-cuts to compute the random MAP perturbations. We compared these bounds to the different forms of structured mean-field, taking the one that performed best: standard structured mean-field that we computed over the vertical chains [8, 1], and the negative tree re-weighted computed on the horizontal and vertical trees [14]. We also compared to the sum-product belief propagation algorithm, which was recently proven to produce lower bounds for attractive models [20, 18]. We computed the error in estimating the logarithm of the partition function, averaged over 10 spin glass models, see Figure 1. One can see that the probable bound is the tightest when considering the medium and high coupling domain, which is traditionally hard for all methods. As it holds in probability it might generate a solution which is not a lower bound. One can also verify that on average this does not happen. The expected lower bound is significantly worse for the low coupling regime, in which many configurations need to be taken into account. It is (surprisingly) effective for the high coupling regime, which is characterized by a few dominant configurations.

Section 4 describes an algorithm that generates unbiased samples from the full Gibbs distribution. Focusing on spin glass models with strong local field potentials, it is well know that one cannot produce unbiased samples from the Gibbs distributions in polynomial time [3]. Theorem 3 connects

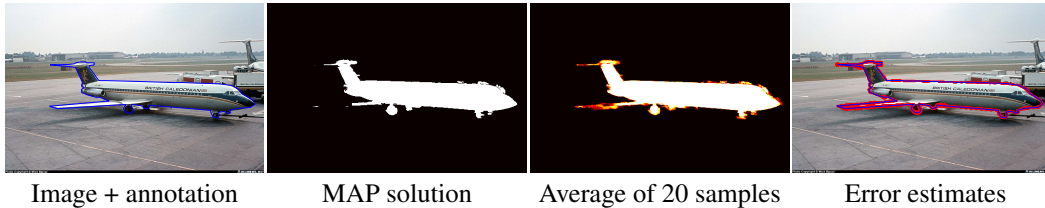

| Image + annotation | MAP solution | Average of 20 samples | Error estimates |

Figure 2: *Example image with the boundary annotation (*left*) and the error estimates obtained using our method (*right*). Thin structures of the object are often lost in a single MAP solution (*middle-left*), which are recovered by averaging the samples (*middle-right*) leading to better error estimates.*

the computational complexity of our unbiased sampling procedure to the gap between the logarithm of the partition function and its upper bound in [5]. We use our probable lower bound to estimate this gap on large grids, for which we cannot compute the partition function exactly. Figure 1 suggests that the running time for this sampling procedure is sub-exponential.

Sampling from the Gibbs distribution in spin glass models with non-zero local field potentials is computationally hard [7, 3]. The approximate sampling technique in Theorem 3 suggests a method to overcome this difficulty by efficiently sampling from a distribution that approximates the Gibbs distribution on its marginal probabilities. Although our theory is only stated for graphs without cycles, it can be readily applied to general graphs, in the same way the (loopy) belief propagation algorithm is applied. For computational reasons we did not expand the graph. Also, we experiment both with pairwise perturbations, as Theorem 2 suggests, and with local perturbations, which are guaranteed to preserve the potential function super-modularity. We computed the local marginal probability errors of our sampling procedure, while comparing to the standard methods of Gibbs sampling, Metropolis and Swendsen-Wang[1]. In our experiments we let them run for at most $1e8$ iterations, see Figure 1. Both Gibbs sampling and the Metropolis algorithm perform similarly (we omit the Gibbs sampler performance for clarity). Although these algorithms as well as the Swendsen-Wang algorithm directly sample from the Gibbs distribution, they typically require exponential running time to succeed on spin glass models. Figure 1 shows that these samplers are worse than our approximate samplers. Although we omit from the plots for clarity, our approximate sampling marginal probabilities compare to those of the sum-product belief propagation and the tree re-weighted belief propagation [22]. Nevertheless, our sampling scheme also provides a probability notion, which lacks in the belief propagation type algorithms. Surprisingly, the approximate sampler that uses pairwise perturbations performs (slightly) worse than the approximate sampler that only use local perturbations. Although this is not explained by our current theory, it is an encouraging observation, since approximate sampler that uses random MAP predictions with local perturbations is orders of magnitude faster.

Lastly, we emphasize the importance of probabilistic reasoning over the current variational methods, such as tree re-weighted belief propagation [22] or max-marginal probabilities [10], that only generate probabilities over small subsets of variables. The task we consider is to obtain pixel accurate boundaries from rough boundaries provided by the user. For example in an image editing application the user may provide an input in the form of a rough polygon and the goal is to refine the boundaries using the information from the gradients in the image. A natural notion of error is the average deviation of the marked boundary from the true boundary of the image. Given a user boundary we set up a graphical model on the pixels using foreground/background models trained from regions well inside/outside the marked boundary. Exact binary labeling can be obtained using the graph-cuts algorithm. From this we can compute the expected error by sampling multiple solutions using random MAP predictors and averaging. On a dataset of 10 images which we carefully annotated to obtain pixel accurate boundaries we find that random MAP perturbations produce significantly more accurate estimates of boundary error compared to a single MAP solution. On average the error estimates obtained using random MAP perturbations is off by 1.04 pixels from the true error (obtained from ground truth) whereas the MAP which is off by 3.51 pixels. Such a measure can be used in an active annotation framework where the users can iteratively fix parts of the boundary that contain errors.

Figure 2 shows an example annotation, the MAP solution, the mean of 20 random MAP solutions, and boundary error estimates.

# 7    Related work

The Gibbs distribution plays a key role in many areas of science, including computer science, statistics and physics. To learn more about its roles in machine learning, as well as its standard samplers, we refer the interested reader to the textbook [11]. Our work is based on max-statistics of collections of random variables. For comprehensive introduction to extreme value statistics we refer the reader to [12].

The Gibbs distribution and its partition function can be realized from the statistics of random MAP perturbations with the Gumbel distribution (see Theorem 1), [12, 17, 21, 5]. Recently, [16, 9, 17, 21, 6] explore the different aspects of random MAP predictions with low dimensional perturbation. [16] describe sampling from the Gaussian distribution with random Gaussian perturbations. [17] show that random MAP predictors with low dimensional perturbations share similar statistics as the Gibbs distribution. [21] describe the Bayesian perspectives of these models and their efficient sampling procedures. [9, 6] consider the generalization properties of such models within PAC-Bayesian theory. In our work we formally relate random MAP perturbations and the Gibbs distribution. Specifically, we describe the case for which the marginal probabilities of random MAP perturbations, with the proper expansion, approximate those of the Gibbs distribution. We also show how to use the statistics of random MAP perturbations to generate unbiased samples from the Gibbs distribution. These probability models generate samples efficiently thorough optimization: they have statistical advantages over purely variational approaches such as tree re-weighted belief propagation [22] or max-marginals [10], and they are faster than standard Gibbs samplers and Markov chain Monte Carlo approaches when MAP prediction is efficient [3, 25]. Other methods that efficiently produce samples include Herding [23] and determinantal processes [13].

Our suggested samplers for the Gibbs distribution are based on low dimensional representation of the partition function, [5]. We augment their results in a few ways. In Lemma 2 we refine their upper bound, to a series of sequentially tighter bounds. Corollary 2 shows that the approximation scheme of [5] is in fact a lower bound that holds in probability. Lower bounds for the partition function have been extensively developed in the recent years within the context of variational methods. Structured mean-field methods are inner-bound methods where a simpler distribution is optimized as an approximation to the posterior in a KL-divergence sense [8, 1, 14]. The difficulty comes from non-convexity of the set of feasible distributions. Surprisingly, [20, 18] have shown that the sum-product belief propagation provides a lower bound to the partition function for super-modular potential functions. This result is based on the four function theorem which considers nonnegative functions over distributive lattices.

# 8    Discussion

This work explores new approaches to sample from the Gibbs distribution. Sampling from the Gibbs distribution is key problem in machine learning. Traditional approaches, such as Gibbs sampling, fail in the "high-signal high-coupling" regime that results in ragged energy landscapes. Following [17, 21], we showed here that one can take advantage of efficient MAP solvers to generate approximate or unbiased samples from the Gibbs distribution, when we randomly perturb the potential function. Since MAP predictions are not affected by ragged energy landscapes, our approach excels in the "high-signal high-coupling" regime. As a by-product to our approach we constructed lower bounds to the partition functions, which are both tighter and faster than the previous approaches in the "high-signal high-coupling" regime.

Our approach is based on random MAP perturbations that estimate the partition functions with expectation. In practice we compute the empirical mean. [15] show that the deviation of the sampled mean from its expectation decays exponentially.

The computational complexity of our approximate sampling procedure is determined by the perturbations dimension. Currently, our theory do not describe the success of the probability model that is based on the maximal argument of perturbed MAP program with local perturbations.

## Footnotes

[1]We used Talya Meltzer's inference package.

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
