[Reviews · NeurIPS 2013]

Submitted by Assigned_Reviewer_5

This paper proposes several approaches to sample from a Gibbs distribution over a discrete space by solving randomly perturbed combinatorial optimization problems (MAP inference) over the same space. The starting point is a known result [5] that allows to do sampling (in principle, using high dimensional perturbations with exponential complexity) by solving a single optimization problem. In this paper they propose to
1) use more efficient low-dimensional random perturbations to do approximate sampling (with probabilistic accuracy guarantees on tree structured models)
2) estimate (conditional) marginals using ratios of partition function estimates, and sequentially sample variables. They propose a clever rejection strategy based on self reduction that guarantees unbiasedness of the samples.
Low-dimensional perturbations are also used heuristically on loopy graphs, showing some promising empirical results on Ising models and for an image segmentation problem. New lower bounds on the partition function are also derived and evaluated on Ising grids, showing improvements over variational bounds when interactions are strong.

Sampling is a fundamental problem with lots of applications, so this paper is definitely very relevant for NIPS. This direction of using random perturbations and MAP inference is as far as I know novel and very different from traditional MCMC sampling. I think there is a potential for this type of approach, especially for probabilistic models with hard dependencies that are notoriously hard for MCMC. I also like the fact that MAP inference is used as a black box, so that future improvements on MAP solvers would directly translate into better sampling methods.

This is an interesting paper making a step forward towards understanding MAP inference with random perturbations, but I feel there is still a significant gap between theory and practice. The theory on low dimensional perturbations (section 3) only works for trees, so it's not really useful in practice (it can be done by dynamic programming). Low dimensional perturbations are then used heuristically on loopy models showing some good empirical results, but without any guarantee (not even asymptotically), which is undesirable. On the other hand the rejection sampler (section 4) is shown to work in theory but not really evaluated in practice.

The paper is generally well written and easy to read. The intuitive idea is clear, and the mathematical formalization is also good except for a few minor things (see below)

The idea of casting the averaging of multiple MAP queries into a single one is clear. Is there an advantage (computational or statistical) in using the extended model to do the averaging in a single run over collecting multiple (independent) samples perturbing the original model and then averaging them (this seems to be what is done in the experiments)?
Since the graphical model is extended creating multiple copies of the variables, which “copy” do you propose to use as a sample?

I like the approach in section 4. The self-reduction and rejection strategy is clever. One issue that is not addressed is how to compute the expectations (line 1 in the pseudocode). Do you have any concentration results on those expectations (like end of section 2)? can you estimate them with high probability using a small number of samples? Wouldn't sample averages introduce some bias (and some failure probability)?

I found the part on lower bounds (section 5) interesting, but not so relevant to the sampling problem addressed in the paper.

Related to this, I would have liked to see more experiments on sampling in place of the ones on the tightness of the bounds on the partition function (a topic already addressed in [6]).
It would have been more convincing to actually run algorithm 1 and plot the runtime (taking into account the time for MAP inference) in place of the second experiment (figure 1, middle). This would also have answered the issue of how the expectations are estimated, and the resulting quality of the samples if the estimates are not exact.

The last two experiments on sampling using low dimensional perturbations are interesting, and show the potential of this type of approach. I’m surprised to see that Gibbs sampling performs so poorly, even for low couplings strength (I would expect the performance of Gibbs sampling to be good for almost uniform probability distributions, and then drop as the coupling strength increases). Any intuition why it is not the case? Perhaps something like Swendsen-Wang would also be a good (stronger) baseline, and a more fair comparison with respect to variational methods.

Other minor comments:

the frequent use of “efficiently” might be misleading considering that it involves solving a potentially NP-hard optimization problem

is theta^hat on line 127-128 a function of all the copies x_{alpha,j_alpha} (extended variables) or just x_alpha?

line 140-141, it would be better to use a different symbol for the extended variable vector (x is already used)

i is not binded in the sum defining theta^hat in line 156. maybe missing a sum? (same thing in corollary 2)

how are the m_i chosen in theorem 2? the restriction m_r=m_s=1 in line 158 is not clear

line 161,what does it mean that x_r,x_s \in argmax? the argmax should be a set of solutions (assignments to the extended variables), while it seems x_r and x_s are single variables

what is |dom(theta)| on line 268? Is it 2^n (for n binary variables)?

line 339, The sub-exponential runtime comment should be made more precise, specifying that the tightness of the bounds (hence rejection probability) is dependent on the graphical model (e.g. coupling strength for Ising grids)

what is the size of the grid used for the approximate sampling experiment? (figure 1 right)

If possible, it would be good to use higher resolution figures as they currently do not print well.
Summary: This paper looks at the idea of sampling by taking solutions of randomly perturbed combinatorial optimization problems (MAP inference). It presents some theoretical analysis and some good empirical results, but there is still a gap between theory and practice.

Submitted by Assigned_Reviewer_7

Authors build on previous work that explores
two uses of MAP inference proposed in (Gumbel Lieblein 1954).
We place authors' contribution in the context of the prior work.

(Gumbel Lieblein 1954) added a high-order perturbation
to the canonical parameters of a distribution from the exponential family
and proved that MAP inference on the perturbed distribution can be used
to sample from the original distribution as well as
to approximate the partition function of the original distribution by a sample mean.

Recent advances in MAP inference led to an interest in
practical low-order perturbations to draw approximate samples.
(Papandreou Yuille 2011) showed how
MAP inference on a low-order perturbed distribution can be used
to sample from the original distribution approximately and
empirically investigated the quality of the approximate samples.

(Hazan Jaakkola 2012) showed that
MAP inference on a low-order perturbed distribution can be used
to lower bound the partition function of the original distribution.

The contribution of the submitted work is twofold.
First the authors propose an efficient approximate sampler and
second the authors propose two lower bounds on the partition function.
In experiments the approximate sampler is first compared with the Gibbs sampler and with the BP
and second the lower bounds are compared with structured mean field and with the BP.


As a side note I believe the authors should cite (Hazan Jaakkola 2012) at ICML
instead of (Hazan Jaakkola 2012) in arXiv.
Summary: Authors build on previous work that explores the use of MAP inference
to approximately sample from an exponential family distribution as well as
to lower bound the partition function of the distribution by a sample mean.

The authors propose an efficient approximate sampler and
two lower bounds on the partition function.

Submitted by Assigned_Reviewer_8

This paper is a follow-up to the recent ICML paper of Hazan & Jaakkola [6],
which used random MAP perturbations to provide bounds on the log
partition function.

The main new contributions are as follows:
- developing random MAP theory for the case of (junction) tree-structured
models, showing that factorized perturbations can be used to draw
"probably approximate samples" from the Gibbs distribution.

- A sequential rejection sampler that uses random MAP as the inner loop.
If the sampler accepts, then the result is a sample from the Gibbs distribution.

- Various other refinements on the theory of random MAP: improved lower bounds
over those in [6] and showing that the approximation scheme of [6] is a lower bound.


Detailed Comments:
- I don't find the descriptions that involve making copies of variables to
be very clear. For example, what does the notation on line 127-129 mean?
\hat theta(x_alpha) = \sum theta(x_{alpha,j_alpha})/m_alpha

How does x_alpha relate to the x_{alpha,j_alpha}'s? What is the dimension
of x_alpha? Are we supposed to assume the later definition of x on line 140/141?
But then what is the difference between x and x_alpha? Why can we use
both x_alpha and x_{alpha,j_alpha} as arguments for gamma_{alpha,j_alpha}
(line 129)?

In general, I think more care needs to go into making these notations
clear, and I'd strongly recommend adding diagrams that support the
mathematical descriptions in Lemma 1 and Theorem 2.

- My impression is that these partition function estimation problems are
difficult near a critical temperature, not that they get monotonically
more difficult as the temperature decreases. Particularly under the assumption
of this work, in which we have access to good MAP routines, this seems
important to be clear about. At very low temperature we can use a single MAP
to get a near perfect estimate. More practically, there are MCMC methods
that are well-equipped to deal with the low temperature regime. For example,
Swendsen-Wang sampling could plausibly make the large moves in configuration
space that would be required for good MCMC sampling from the Gibbs distribution.
So I don't find the fact that results are good in the high coupling case
to be as compelling as the paper makes them out to be. I'd like to see
a comparison to Swendsen-Wang as well as the simple baseline of estimating
the partition function using a single unperturbed MAP.

- The abstract sounds really similar to [6]. It'd be nice to emphasize the
differences more.
Summary: Overall, there are a number of interesting results that help flesh out the
theory of random MAP perturbations. There are some issues with clarity in
the descriptions (see detailed comments), and more sophisticated MCMC methods should be considered as alternatives, but despite some reservations, I generally found
this to be a good and interesting paper.
Author Feedback

Author rebuttal: We thank the reviewers for their effort and comments that are helping to improve our paper.

Sampling from the Gibbs distribution is a notoriously hard task when there are arbitrary external fields, i.e., different data terms for every variable. This setting is unique to machine learning applications (where data is important) and rarely studied in mathematics/statistics (which consider a pure interaction models, without data). The presence of external fields can make the probability landscape very ragged, even in the case of attractive interaction potentials. Due to the external fields, weak interactions between the variables do not directly translate into uniform distribution. Similarly, while we might expect likely assignments to concentrate around clusters with strong coupling strengths, this picture may be destroyed by strong external fields. The high signal and high coupling regime is very important to machine learning applications, as these applications are governed by data (high signal) and knowledge (high coupling). The MCMC methods, including Swendsen-Wang, fail in the presence of external fields, e.g., [4] for theoretical explanation and Barbu 2007 for practical evidence with the Swendsen-Wang. The success of the Swendsen-Wang (e.g, Cooper 1998, Huber 2002, Ullrich 2011) is limited to models without external fields, a case which can be solved in closed-form for grids by the Pfaffian method (Kasteleyn 1963). We will compare to the Swendsen-Wang in the final submission. The lower bound baseline of the MAP is worse than our lower bounds, as well as the standard structured-mean-field and belief propagation which we outperform - these standard lower bounds are both known to be better than the MAP bound (which is essentially the bound of [6]), see [6] page 6 for a direct reference.

Assigned_Reviewer_5
1) We agree that there is a gap between theory and practice. This is akin to belief propagation, where theory was initially available only for trees. Our contribution is a similar step towards bridging the gap.
2) In Lemma 1 there is no difference between using a single expanded MAP and averaging multiple MAP queries. In Theorem 2 we compress exponential number of MAP queries to a single one, using graph separations. Any copy can be used as a sample.
3) Concentration results for Section 4 can be derived, since the variance of the maximum is bounded (e.g., by the sums of variances). Sample averages will introduce approximate samples.
4) theta^hat on line 127-128 a function of all the copies x_{alpha,j_alpha}
5) m_i is chosen to determine the approximation quality. Since m_r=m_s=1 it is a single variable. It can be extended to any m_r=m_s and in this case x_r,x_s may be any of the copies.
6) the domain is the set of x \in X for which theta(x) > -\infty. In spin glass models that we used it is 2^n.
7) The experiment for the approximate sampling considers 10x10 grid.

Assigned_Reviewer_8
1) theta^hat on line 127-128 a function of all the copies x_{alpha,j_alpha}. In retrospect we should have used different symbols to x_alpha and x_{alpha,j_alpha}. We will clarify this and related notations. Thanks.
2) Max-product with Gumbel perturbations gives the sum product. This in turn gives marginal probability distributions (upon convergence) that can be sampled from, independent of the algorithm. Theorem 2 produces approximate samples with a single MAP.
3) The partition function is hard whenever there are external fields (or equivalently, local data terms). This nature of its hardness was not extensively explored in statistics/mathematics/computer science yet. For lower bounds, the single MAP baseline is worse than the standard methods of structured mean-field and belief propagation, which we outperform. Please refer to the beginning of the rebuttal for more information.